# Scaling Laws and Complexity of Generative Models: A Multifractal Perspective

## Abstract

Assessing the functional aspects of a generative model (GM) is crucial to technological advancement. However, existing evaluation metrics are often insensitive to distributional changes and rarely correlate with perceptual fidelity. In addition, their oversimplified assumptions limit their ability to assess morphological fidelity and contextuality in application areas such as medical imaging and industrial machine vision. Hence, domain-agnostic, robust, and reliable GM evaluation remains an unresolved problem in generative AI and is an ongoing research paradigm. So, this work introduces the concept of multifractality, a scaling technique adapted from statistical physics for GM evaluation. Several multifractal markers are proposed as new metrics to analyze the scaling behavior of GMs. They characterize the structural complexity of long-range correlation patterns in GM-generated images. Non-parametric statistics-based hypothesis testing is formulated to assess the disparity in morphological organization between synthesized and actual data. These metrics are extensively validated using benchmark GMs on real-world datasets. Furthermore, multifractal spectrum analysis provides deeper insights into a GM's complexity origin and plausible spectral bias explainability.

## 1 Introduction

Contemporary GMs, such as *generative adversarial networks* (GANs) Goodfellow et al. (2020); Karras et al. (2019; 2020; 2021), *variational autoencoders* (VAEs) Kingma & Welling (2013); Razavi et al. (2019), *normalizing flows* (NF) Kingma & Dhariwal (2018), and *diffusion models* Ho et al. (2020b); Song et al. (2020); Kingma et al. (2021); Song et al. (2021); Dhariwal & Nichol (2021); Vahdat et al. (2021) have demonstrated unprecedented fidelity, often exceeding human evaluative capacities Dhariwal & Nichol (2021); Rombach et al. (2022). Consequently, this has raised questions on the sufficiency of existing evaluation metrics to the extent of how far these GM architectures truly learned the ground truth distribution. Furthermore, it has been debated whether striving for cutting-edge performance using these GMs provides appropriate research objectives to drive future algorithmic progress in generative artificial intelligence (GenAI) Stein et al. (2024).

Unlike discriminative models $P(T|X)$, which can be easily evaluated using a few labelled samples $(X_i; T_i)$, GM evaluation remains challenging. Although the likelihood function of a GM appears intuitive for assessment, it is generally intractable and scales poorly in higher dimensions. The GM evaluation literature abounds in metrics such as Inception score, Fréchet Inception distance, Kernel-Inception distance and a variety of k-nearest neighbour (KNN)-based two-value metrics, *precision* and *recall* (P&R) Sajjadi et al. (2018); Simon et al. (2019); Kynkäänniemi et al. (2019); Naeem et al. (2020); Alaa et al. (2022), improved P&R Kynkäänniemi et al. (2019), probabilistic Park & Kim (2023), and density and coverage Naeem et al. (2020) for GM evaluation. These metrics quantify the statistical divergence between the real $\{X_i\}$ and generated (fake) $\{Y_i\}$ samples using lower-dimensional representations. These are computed using a *Inception-V3* model trained on *ImageNet* dataset Razavi et al. (2019); Karras et al. (2020); Dhariwal & Nichol (2021); Zheng et al. (2021); Anokhin et al. (2021); Chan et al. (2021); Ntavelis et al. (2022). However, their over-reliance on *Inception-V3* embeddings makes them agnostic to features not related to the 1k classes of *ImageNet* Heusel et al. (2017); Kynkäänniemi et al. (2019); Naeem et al. (2020). In addition, these metrics are insensitive to fake distributional changes, unreliable due to outlier susceptibility, and rarely correlate with human perceptual scores Zhou et al. (2019); Stein et al. (2024). The absence of a

**Figure 1:** Schematic of multifractal analysis-based GM evaluation pipeline.

direct correlation with contextual morphological fidelity limits their practical applicability in critical scenarios such as medical imaging Aggarwal et al. (2021); Springenberg et al. (2023).

Medical imaging typically encodes multi-scale and non-linear micro-structural tissue convolutions with rich morphological diversity Das et al. (2014); Nawn et al. (2020). Similarly, industrial machine vision applications, such as hydrophobicity dielectrics or gas-insulated switchgear, include complex, multi-scale and non-linear spatial heterogeneity patterns Tang et al. (2015); Wang et al. (2025). Accordingly, GM-generated images must capture these task-specific fine-grained feature representations, which are arguably one of the most critical bottlenecks in GenAI applications Lopes & Betrouni (2009); Das et al. (2014); Nawn et al. (2020); Islam et al. (2013); Wu et al. (2023). Although prevalent GMs can produce photorealistic natural images, their efficacy in real-world medical imaging and industrial contexts remains challenging. In addition, standard GM evaluation metrics have yet to provide a clear consensus for these applications Yi et al. (2019); Wu et al. (2023). So, this work addresses the scientific question, *'To what extent do current GM architectures generate samples consistent with the morphological organization and complexity of real-world data?'*

Beyond the spatial heterogeneity in GM-generated data, the origin of the *spectral bias* phenomenon in deep neural networks (DNNs) is not fully understood using the prevailing approaches, and still eludes researchers Fridovich-Keil et al. (2022). *spectral bias* is the propensity of DNNs to learn simple solutions for highly expressive functions that generalize surprisingly well. Consequently, DNNs are known to produce samples with elevated frequencies Rahaman et al. (2019); Schwarz et al. (2021). Thus, developing new techniques underpinning novel *structural* and *functional* aspects of a GM is essential for accelerating progress in generative AI Alaa et al. (2022). An active research area has been examining the dynamics of GM-generated data from a complexity science paradigm. These include image-space motion trajectories Li et al. (2024) and scaling laws in natural languages De Santis et al. (2023; 2024). It promises immense potential for applying sophisticated signal processing techniques for GM evaluation. Within this context, the concept of GM's *multifractality* is introduced. Multifractal analysis is a *scaling* technique adapted from *statistical physics*, to characterize the power-law scaling dynamics. Although *power-law scaling* has been applied in medical imaging, primarily for representation learning Stosic & Stosic (2006); Lopes & Betrouni (2009); Islam et al. (2013); Das et al. (2014); Nawn et al. (2020), its potency for GM's fidelity evaluation has not been explored to date and forms the primary motivation for this work.

Specifically, we develop a parametric evaluation framework for GM evaluation. Multiple statistically significant parameters are introduced in terms of multifractality and long-range spatial correlations. These parameters describe the morphological disparity between the synthesized and actual images. They also provide deeper insights into the complexity of generative structures. As intuited by non-linear dynamics, GM-generated images could be evaluated using the Hurst exponent and singularity spectrum width markers. To summarize, our main contributions are enumerated as:

- Proposing an information-theoretic framework (shown in Figure 1) to evaluate the fidelity of GM. The method uses multifractal analysis of GM-generated images. It proposes multiple metrics for the evaluation of GM in terms of complexity and long-range spatial correlation patterns to characterize their spatial dynamics according to the power law.

- Providing theoretical and numerical support for the evaluation metrics through experiments on real-world datasets with benchmark GM architectures. Our findings characterize the morphological organization and complexity of GM-generated images. It also provides deeper insights into *spectral bias* explainability.

## 2 PROPOSED METHODOLOGY AND PRELIMINARIES

Figure 1 outlines a detailed schematic of the proposed GM evaluation framework. The pipeline integrates the following modules: 2D-Multifractal Detrended Fluctuation Analysis (2D-MFDFA), extraction of multifractal markers from GM-generated synthesized images, their exploratory analysis, and hypothesis testing-based formulation of fidelity metrics. We briefly discuss the technical intricacies of different modules in the block diagram as follows.

### 2.1 2D-MFDFA

2D-MFDFA is a technique to examine the power-law scaling dynamics of a surface or image in terms of long-range spatial correlations and complexity of morphological structures Kantelhardt et al. (2002); Xi et al. (2016). An image is represented by a two-dimensional array, $X(i,j)$, where $i = 1, 2, \cdots, N_1$ and $j = 1, 2, \cdots, N_2$. 2D-MFDFA includes the following steps:

**Step 1**: $X(i,j)$, is divided into $N_{1s} \times N_{2s}$ disjointed square segments of size $s \times s$, where $N_{1s} = [N_1/s]$ and $N_{2s} = [N_2/s]$. These segments are represented as $X_{v,w} = X_{v,w}(i,j) = X(l_1+i, l_2+j)$ $\forall 1 \leq i, j \leq s$, where $l_1 = (v-1)s$, $v = 1, 2, \cdots, N_{1s}$, $l_2 = (w-1)s$, and $w = 1, 2, \cdots, N_{2s}$.

**Step 2:** In each of these segments $X_{v,w}$, we compute a cumulative sum $u_{v,w}(i,j)$ as:

$$u_{v,w}(i,j) = \sum_{k_1=1}^{i} \sum_{k_2=1}^{j} X_{u,w}(k_1, k_2) \tag{1}$$

where $i, j = 1, 2, \cdot, \cdot, \cdot, s$. Note that $u_{v,w}$ itself is a surface.

**Step 3:** The trend in the constructed surface, $u_{v,w}$ is computed by fitting a pre-determined bi-variate polynomial function $\tilde{u}$. Generally, a simple linear detrending function is used for detrending as:

$$\tilde{u}_u, w(i,j) = ai + bj + c, \tag{2}$$

where $1 \leq i, j \leq s$ and a, b and c are polynomial parameters to be estimated. Eqn. 2 denotes a plane characterized by simple matrix operations in the least-squares sense. Accordingly, the residual matrix, $\varepsilon_{i,j}$ is computed as:

$$\varepsilon_{i,j}(i,j) = u_{v,w}(i,j) - \tilde{u}_{v,w}(i,j) \tag{3}$$

The sample variance of the residual matrix $\varepsilon_{v,w}$ is used to compute the fluctuation function $F_{v,w}(s)$ for segment $X_{v,w}$

$$F_{v,w} = \sqrt{\frac{1}{s^2} \sum_{i=1}^{s} \sum_{j=1}^{s} (\varepsilon_{v,w}(i,j))^2} \tag{4}$$

**Step 4:** The overall detrended fluctuation function, $F_q(s)$ is computed by taking the mean of $F_{v,w}$ across all the segments and is calculated as:

$$F_q(s) = \left\{ \frac{1}{N_{1s}N_{2s}} \sum_{u=1}^{N_{1s}} \sum_{w=1}^{N_{2s}} F_{v,w}^q(s) \right\}^{\frac{1}{q}} \tag{5}$$

Where $q$ can take any real value except $q = 0$. At $q = 0$, the factor $\frac{1}{q}$ in Eqn. 5 blows up, so normal averaging fails. Instead, a logarithmic averaging procedure using L'Hôspital's rule is applied to compute $F_0(s)$, and is calculated as:

$$F_o(s) = exp \left\{ \frac{1}{N_{1s}N_{2s}} \sum_{u=1}^{N_{1s}} \sum_{w=1}^{N_{2s}} ln[F_{u,w}(s)] \right\} \tag{6}$$

,

**Step 5:** We vary the scale size, $s$ in $[s_{min}, s_{max}]$ where $s_{min} = 10$ to $s_{max} = min[N1, N2]/4$ and compute the power-law scaling relationship between $F_q(s)$ and $s$,

$$F_q(s) \sim s^{h(q)} \tag{7}$$

A surface exhibiting power-law scaling behaviour will have $\log F_q$ linearly depending on $\log s$, with $h(q)$ as the slope. Here, $h(q)$ is the Generalized Hurst Exponent. $h(q)$ characterize the scaling nature of the segments with large (small) fluctuations for positive (negative) exponent values of $q$. The value of $h(q = 2)$ is known as the Hurst Exponent, which characterizes a surface's long-range persistence or memory. It is related to the decay rate of the statistical dependence between two points on a surface with an increasing gap between them Ashkenazy et al. (2002). Moreover, $h(q)$ is related to the classical scaling exponent, $\tau(q)$ as:

$$\tau(q) = qh(q) - D_f \tag{8}$$

where $D_f$ is the fractal dimension of the geometric support of the multifractal measure and $D_f = 2$ for a surface.

$h(q)$ is independent of $q$ and $\tau(q)$ is linearly dependent on $q$ for a monofractal surface. So, a unique value of $h(q)$ characterizes the spatial dynamics of a monofractal surface Kantelhardt et al. (2002). In contrast, $h(q)$ is a function of $q$, and $\tau(q)$ varies non-linearly with $q$ for a multifractal surface Ashkenazy et al. (2003). Moreover, the singularity spectrum, $f(\alpha)$ for a multifractal surface is related to $\tau(q)$ by Legendre transform Shao & Ditlevsen (2016), and is given by:

$$\alpha = d\tau(q)/dq, \qquad f(\alpha) = q\alpha - \tau(q) \tag{9}$$

Where $\alpha$ is the *Hölder exponent* and $f(\alpha)$ is the multifractal spectrum. $f(\alpha)$ characterizes the relative relative strength of the range of fractal exponents ($\alpha$) to the analyzed surface Shimizu et al. (2002). Using Eqn. 8, we can write Eqn. 9 as:

$$\alpha = h(q) + qh'(q), \qquad f(\alpha) = q[\alpha - h(q)] + 2 \tag{10}$$

Usually, a quadratic function is fitted to the multifractal spectra around the position of maximum $\alpha_0$ for analysis as:

$$f(\alpha) = A(\alpha - \alpha_0)^2 + B(\alpha - \alpha_0) + C \tag{11}$$

Where $C$ is an additive constant, whose value is given by $C = f(\alpha_0)$. Here, $B$ signifies the singularity spectrum asymmetry and is zero for a symmetric spectrum, while $A$ indicates the concavity ($A < 0$) or convexity ($A > 0$) of the singularity spectrum. The singularity spectrum width, $\Delta\alpha$ of the singularity spectrum is computed by extrapolating Eqn. 11 to zero. $\Delta\alpha$ is computed as:

$$\Delta\alpha = \alpha_{max} - \alpha_{min}, \quad s.t. \ f(\alpha_{max}) = f(\alpha_{min}) = 0 \tag{12}$$

Here, $\alpha_{max}$ and $\alpha_{min}$ correspond to the right and left extremities of $f(\alpha)$ on the f($\alpha$) vs. $\alpha$ plot. The value of $\Delta\alpha$ characterizes the degree of multifractality of an image and correlates with its singularity strength Shimizu et al. (2002). For a monofractal surface, $h(q)$ is independent of $q$, which is equivalent to a unique value of $f(\alpha)$ describing the multifractal spectrum $\forall \alpha$; hence, $\Delta\alpha = 0$ (following Eqn. 10 and 12).

## 2.2 MULTIFRACTAL MARKERS FOR FIDELITY EVALUATION

Figure 2 demonstrates a typical f($\alpha$) vs. $\alpha$ plot exemplifying the multifractal spectrum and different shape parameters derived from its geometry. These parameters characterize the morphological organization of an image in terms of the complexity of morphological structures Tang et al. (2015); Nawn et al. (2020). The significance of different parameters extracted from 2D-MFDFA for fidelity evaluation is described in Table 1.

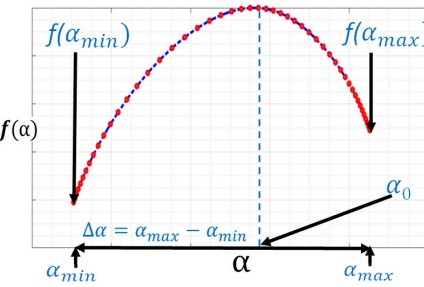

**Figure 2:** Illustration of different shape parameters extracted from a typical multifractal spectrum, i.e., plot of f($\alpha$) vs. $\alpha$.

| Marker | Description |
|---|---|
| $h(q = 2)$ | Indicate the nature of long-range spatial correlations. |
| $\alpha_{min}$ | Right extremity of the singularity exponent. |
| $\alpha_{max}$ | Left extremity of the singularity exponent. |
| $f(\alpha_{min})$ | Singularity spectrum value at $\alpha_{min}$. |
| $f(\alpha_{max})$ | Singularity spectrum value at $\alpha_{max}$. |
| $\Delta\alpha$ | Singularity strength. Indicate the degree of multifractality. |
| $\alpha_0$ | Location of Singularity spectrum's maxima/minima. |
| $f(\alpha_0)$ | Singularity spectrum value at $\alpha_0$. |
| A | Singularity spectrum's concavity ($A < 0$) or convexity ($A > 0$). |
| B | Singularity spectrum asymmetry. |

**Table 1:** Description of extracted multifractal markers.

All the multifractal markers described in Table 1 are used to evaluate the fidelity of the GM-generated images. Here, $h(q = 2)$ characterizes the long-range spatial correlations in an image. $\alpha_{min}$ and $\alpha_{max}$ exemplify the bounds of the singularity exponent, $\alpha$ in the 2D-MFDFA investigation Nawn et al. (2020). $f(\alpha_{min})$ and $f(\alpha_{max})$ indicate the singularity spectrum value at the left and right extremities of the singularity exponent $\alpha$ in f($\alpha$) vs. $\alpha$ plot. $\Delta\alpha$ indicates an image's singularity strength computed between the extremities of $\alpha$. $\Delta\alpha = 0$ for a monofractal surface, and larger values correlate with a higher degree of multifractality. A higher value of $\alpha_0$ signifies an image's highly coherent and regular morphological structures. $f(\alpha_0)$ indicates the maximum strength of the singularity exponent and is a measure of roughness in an image. The singularity spectrum asymmetry index, B (Eqn. 11) describes the asymmetry of the multifractal spectrum geometry. A positive B is associated with lower-order fractal exponents (i.e., the dominance of positive q values), whereas a negative B indicates the dominance of negative q values. Except for $h(q = 2)$ in Table 1, the multifractal spectrum geometry parameters assess an image's multifaceted aspect of morphological organization. The fidelity of the GM-generated synthesized images is evaluated in terms of derived multifractal markers' disparity from real-world images. Moreover, to relax the oversimplified assumptions, we use hypothesis testing-based assessment of the multifractal markers in a probabilistic sense.

### 2.3 EXPLORATORY FEATURE ANALYSIS AND HYPOTHESIS TESTS

We use BoxPlot representation of the derived multifractal markers for visualization of summary statistics (locality, spread, and skewness) Frigge et al. (1989) and use non-parametric statistics-based hypothesis tests Fagerland (2012) for inferential statistical analysis. Non-parametric hypothesis tests have been used to avoid making assumptions about the underlying data distribution. In particular, we use the Wilcoxon rank sum test Woolson (2005) to compare the location (median; $\eta$) of the multifractal markers across the study groups ($G_i$) for different datasets and compute the statistical significance of the null hypothesis: $\eta_{G_i} = \eta_{G_j}$, against the alternative hypothesis: $(\eta_{G_i} < \eta_{G_j})$ $\vee (\eta_{G_i} > \eta_{G_j})$ at a standard significance level of 0.5% Hosmer & Lemeshow (1992). Here, $G_i$ corresponds to the various GM architectures explored in the experimental evaluation. We demonstrate the sensitivity of statistically significant evaluation metrics on the BoxPlots by using $- * -$.

## 3 EXPERIMENTAL EVALUATION

Amidst various imaging modalities, multi-scale morphological structures are noteworthy in the microscopic domain Kaji et al. (2024); Qian et al. (2012); Braumann et al. (2005); Das et al. (2014); Nawn et al. (2020). So, we verified the efficacy of our proposed GM evaluation approach on three real-world challenging histopathological datasets (*BreakHis*, *BACH*, and *Oral Cancer*) and across benchmark GM architecture, namely, *GAN*, *VAE*, *NF*, and *Diffusion*.

**Datasets.** *BreakHis* contain 9,109 (2,480 benign and 5,429 malignant) microscopic images of breast tumour tissues collected from 82 patients at magnifying factors of 40X, 100X, 200X, and 400X Spanhol et al. (2015). *BACH* include 400+ whole-slide microscopy images from four classes, namely, normal, benign, in situ carcinoma and invasive carcinoma Aresta et al. (2019); Araújo et al. (2017); Fondón et al. (2018). *Oral Cancer* comprise 1224 (290 normal epithelium and 934 oral squamous cell carcinoma) histopathological images at 100X and 400X magnification levels Rahman et al. (2020).

**Baseline GMs.** We have used a deep convolutional GAN Radford et al. (2016) to generate the GAN network's images. The generator in the GAN architecture consisted of five upsampling blocks with transposed convolutions, while the discriminator had four downsampling blocks with normal convolutions. Similarly, we used the VAE model in Razavi et al. (2019) to generate the VAE network's images. In our experiments with NF, we used a Flow++ Ho et al. (2019) model with two dequantization blocks and ten residual blocks and used checkerboard coupling layers. The NF model was trained using an Adam Optimizer with a negative log-likelihood loss function. We used a denoising diffusion probabilistic model (DDPM) Ho et al. (2020a) to generate the diffusion-based GM images. 2D UNet from huggingface diffusers[1] has been used as the underlying generator network consisting of six downsampling blocks and restrained upsampling blocks in the DDPM architecture,

---

[1] https://huggingface.co/

with the downsampling and upsampling blocks includes a spatial self-attention mechanism. The DDPM was trained using an Adam Optimizer and mean square error loss function.

**Implementation Details.** An image resolution of $128 \times 128$ was used for training and inference across all the three datasets. Since histopathology images encode intricate cellular architecture of tissues, standard data augmentation through rotation or shearing may distort the anatomical structures, leading to unrealistic variations and histological inconsistency Wei et al. (2019). So, only random horizontal and vertical flips were used to augment the dataset size. No pre-training was performed on the generator model used to generate the images. All the experiments were done on a single 80 GB Nvidia A100 GPU with FP16 precision, and all the GMs were trained with a maximum number of training epochs set to 500. For all the experiments, we have used AdamW optimizer ($\alpha = 0.0001, \beta 1 = 0.9, \beta 2 = 0.999, \epsilon = 10^{-8}$) with a weight decay factor of 0.01. A batch size of 128 and 512 were used during training and data generation, respectively. We generate 2000 images for each dataset with resolution $128 \times 128$, the same as the input images.

## 3.1 EVIDENCE OF POWER-LAW SCALING DYNAMICS.

Figure 3 illustrates the typical variation of $\log F_q(s)$ vs. $\log s$ for different moments ($q$), obtained from 2D-MFDFA on *Original* images taken from Breakhis (Figure 3a), BACH (Figure 3b), and Oral Cancer (Figure 3c) datasets, respectively and the corresponding synthetic images generated by different GM architectures. Log $F_q$ linearly depends on log $s$, with $h(q)$ as the slope, which signifies the power-law scaling behaviour. Moreover, these scaling patterns are consistently observed across all the *Original* images and those generated by different GM architectures, thereby suggesting further analysis of the generalized Hurst exponent, $h(q)$ (Eqn. 7) and singularity spectrum, $f(\alpha)$ vs. $\alpha$ (Eqn. 12) for characterizing the GM-generated image dynamics.

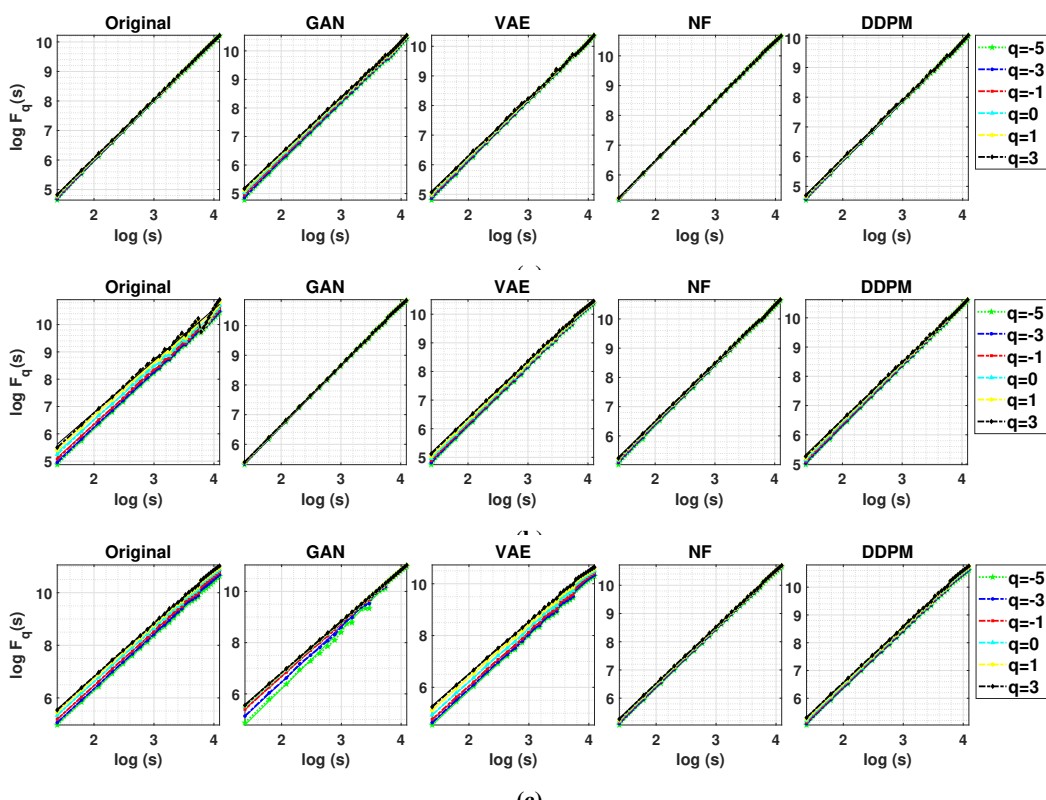

**Figure 3:** Typical variation of $\log F_q(s)$ vs. $\log s$ for different $q$, obtained using 2d-MFDFA of the *Original* image samples and synthetic images generated by different GM architectures. The GMs are trained on (a) Breakhis, (b) BACH, and (c) Oral Cancer datasets, respectively. Here, $\log F_q$ linearly depending on $\log s$, with $h(q)$ as the slope, indicates power-law scaling behaviour.

The typical variation of $\log F_q(s)$ vs. $\log s$ for different moments $(q)$, obtained in Figure 3a for the Breakhis dataset is uniform across all the *Original* images and those generated by different GM architectures, with GAN showing slight $(q)$-dependent slope variations. Similarly, the *Original* images from the BACH dataset exhibits $(q)$-dependent slope variations in the $\log F_q(s)$ vs. $\log s$ plots, which is comparatively less discernible in different GM-generated images in Figure 3b. Lastly, for the Oral Cancer dataset in Figure 3c, the *Original* image exhibits $(q)$-dependent slope variations in the $\log F_q(s)$ vs. $\log s$ plots, which is reciprocated in different GM-generated images.

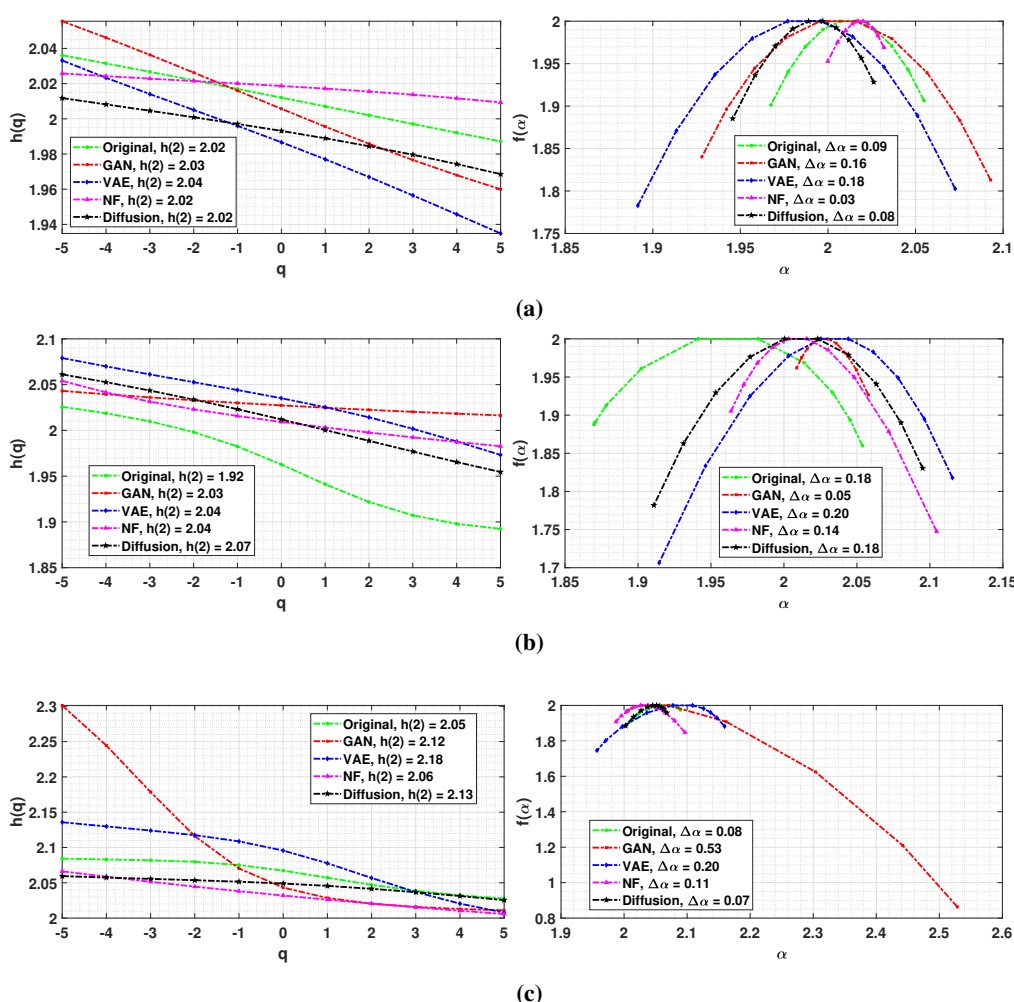

**Figure 4:** Typical $(q)$-dependent generalized Hurst exponent $h(q)$ and singularity spectrum $f(\alpha)$ vs. $\alpha$ is shown for *Original* and GM-generated samples generated from 2D-MFDFA of images, taken from (a) Breakhis, (b) BACH, and (c) Oral Cancer datasets, respectively. Here, the multifractality in the *Original* and GM-generated synthesized images is evident from non-linear $h(q)$ vs. $q$ and $\Delta\alpha \neq 0$.

## 3.2 EVIDENCE OF MULTIFRACTALITY FROM $h(q)$ VS. $q$ AND $f(\alpha)$ VS. $\alpha$ ANALYSIS

Figure 4 demonstrate the typical $q$-dependent $h(q)$ and $f(\alpha)$ vs. $\alpha$ variations, obtained from 2D-MFDFA on *Original* images taken from Breakhis (Figure 4a), BACH (Figure 4b), and Oral Cancer (Figure 4c) datasets, respectively and the corresponding synthetic images generated by different GM architectures. Our results highlight decreasing $h(q)$ with increasing $q$ and $h(q)$ varying non-linearly for negative and positive orders of $q$. This non-linear variation of $h(q)$ with $q$ and $\Delta\alpha \neq 0$ (Refer to Eqn. 12) patterns are consistently observed across all the *Original* images and synthesized images

generated by different GM architectures, thereby highlighting their underlying multifractal nature. Also, the typical values of $h(q = 2) \approx 2$ suggest the presence of long-range spatial correlation patterns, which are persistent for both the *Original* and synthetic images generated by different GM architectures. $\Delta\alpha \neq 0$ from $f(\alpha)$ vs. $\alpha$ variations suggest the analysis of different sources of morphological complexity in an image Zhou et al. (2008).

Figure 5 demonstrates the BoxPlot-based visualization of the derived multifractal markers, $h(q = 2)$, $\alpha_{min}$, $\alpha_{max}$, $f(\alpha_{min})$, $f(\alpha_{max})$, $\Delta\alpha$, and the singularity spectrum asymmetry index, B (Eqn. 11), obtained from 2D-MFDFA of *Original* images taken from the Oral Cancer datasets and synthesized images. Across different GM architectures, our study shows that DDPM-based diffusion models come in close agreement with the *Original* images in terms of the underlying complexity of the morphological structures. In the next section, we study the different sources of complexity and quantify the fidelity in terms of different multifractal markers.

### 3.3 SOURCES OF MORPHOLOGICAL COMPLEXITY

Figure 6 demonstrate the BoxPlot-based visualization of and different sources of complexity from $\Delta\alpha$, $\Delta\alpha_{Shuffled}$, and $\Delta\alpha_{Density}$, obtained from 2D-MFDFA of GM-generated (Diffusion & GAN) and real-world (Original) images taken from Breakhis ($1^{st}$ row), BACH ($2^{nd}$ row), and Oral Cancer ($3^{rd}$ row) datasets, respectively. Here, $\Delta\alpha_{Density}$ is the singularity spectrum width after randomly shuffling all the data points in an image surface and analyzing the multifractality. Shuffling destroys all the long-range correlations, and the resulting source of multifractality is due to the data distribution. $\Delta\alpha_{Shuffled}$ is the difference between the singularity spectrum width obtained from the shuffled and original image. We argue that the spectral bias phenomenon in DNNs is due to the data distribution

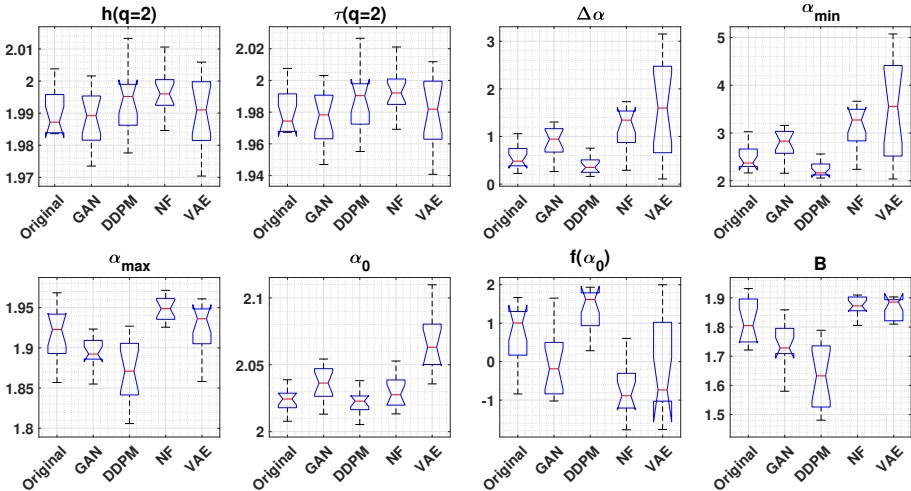

**Figure 5:** BoxPlots showing the variation (locality, spread, and skewness) of the derived multifractal markers, $h(q = 2)$, $\alpha_{min}$, $\alpha_{max}$, $f(\alpha_{min})$, $f(\alpha_{max})$, $\Delta\alpha$, and singularity spectrum asymmetry index, B (Eqn. 11), obtained from 2D-MFDFA of *Original* images taken from the Oral Cancer datasets and synthesized images using different GM architectures.

## 4 CONCLUSION

This work proposed a framework for systematically assessing the fidelity of GM-generated images through the lens of multifractality. Multiple multifractal markers are introduced using non-parametric hypothesis testing for GM evaluation. These metrics characterize long-range correlated dynamics and structural complexity in GM-generated images. They quantify the statistical disparity in the morphological organization between the synthesized and actual data. Experimental results reveal consistent multifractal patterns across GM architectures and real-world datasets. Our analysis provides deeper insight into the origins of the underlying complexity of GM and offers a plausible explanation

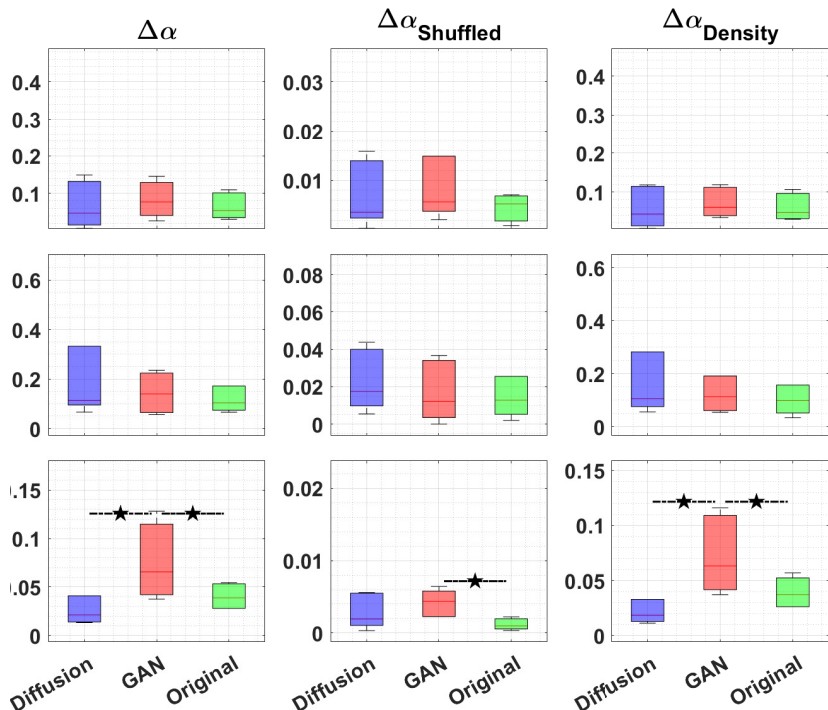

**Figure 6:** BoxPlots showing the variation (locality, spread, and skewness) of the derived multifractal markers ($h(q=2)$, $\Delta\alpha$, $\Delta\alpha_{Shuffled}$, and $\Delta\alpha_{Density}$), obtained from 2D-MFDFA of GM-generated (Diffusion & GAN) and real-world (Original) HI for Breakhis, BACH, and Oral Cancer datasets, respectively. The inter-class statistical difference using the non-parametric Wilcoxon rank sum test at a 0.5% significance level is shown with $-*-$.

for the spectral bias phenomenon. We believe that the proposed metrics will primarily benefit medical imaging research. They enable the fairness assessment of GM-generated histopathological images, where multi-scale deformation dynamics of multifarious micro-structural tissue convolutions are encoded due to long-range spatial correlations. These patterns alter during pathological processes in patient subgroups. We leave an in-depth study of this use case for future work. We also plan to use information-theoretic measures, such as recurrence quantification, for GM evaluation. In addition, our multifractal formalism can easily be extended to 1D and high-dimensional settings.

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
