# OpenReview forum: "Scaling Laws and Complexity of Generative Models: A Multifractal Perspective"
_ICLR.cc/2026/Conference — Submitted to ICLR 2026_

### Official Review · Reviewer_cR6w · 2025-10-20

**Soundness:** 2
**Presentation:** 3
**Contribution:** 2
**Rating:** 2
**Confidence:** 3

**Summary:**

The paper compares the value of a statistic called MFDFA (for multifractal detrended fluctuation analysis) on (1) images from three medical scan datasets and (2) images produced by generative models that are trained on these datasets.

MFDFA is supposed to measure long-range correlations within a signal. The concept makes more sense for time-varying signals but it has been tested on images before revealing a "scaling law". Experiments here confirm that the scaling law holds for both original and generated images with similar scaling exponent.

Additional experiments are carried out to quantify the small differences in the scaling exponents. They are not conclusive.

Lastly the experiment is repeated not on the original images but on random images in which the pixels are randomly permuted (indepenently in each image I suppose)? This supposedly amplifies the scaling exponent differences.

**Strengths:**

The paper convincingly demonstrates that the scaling law present in the datasets is preserved by the generative models that are studied.

**Weaknesses:**

First it is unclear what the thesis of this paper is. Is the claim that original and generated images are distinguishable by your metric? Or that they are indistinguishable?

Whatever the case, no explanation of the experiments is proposed. The conclusion makes several claims that do not appear substantiated. (See questions below.)

The experiments carried out in this paper are an important first step towards formulating a hypothesis about the distinguishability of real and generated images with respect to the "fractality" measures under consideration. Based on your experiments the most plausible hypothesis is the null hypothesis, namely that real and generated images are indistinguishable by your measures. This is not particularly surprising as these models are train to produce images that do appear realistic.

For the work to have value to the scientific community, a thesis based on these experiments ought to be formulated and supported. If there are no differences it is expected that the authors should provide an explanation. For example: do the objective optimized in training implicitly also optimize for MFDFA? Is there an possibly synthetic illustrative model of the phenomenon? If there are differences where do they originate from? Unless there is an interesting *explanation* to be communicated about the result of these experiments I do not see much benefit into including this work in the program.

**Questions:**

Line 431: "Our analysis provides deeper insight into the origins of the underlying complexity of GM and offers a plausible explanation for the spectral bias phenomenon." What is your plausible explanation?

Line 430: "Experimental results reveal consistent multifractal patterns across GM architectures and real-world datasets.".  Your figure 4 graphs have a good linear fit for q > 0. Based on your introduction this is evidence of single-fractal not multi-fractal patterns, no?

---

### Official Review · Reviewer_fWup · 2025-10-30

**Soundness:** 1
**Presentation:** 2
**Contribution:** 1
**Rating:** 2
**Confidence:** 4

**Summary:**

The paper studies the evaluation of generative models and proposes scores for assessing the fidelity and contextuality in medical imaging applications. The method is explained in Section 2, where they use 2D-MFDFA in Section 2.1 by defining $F_q(s)$ in Eq. (5,6) and then describing the power-law scaling in (7). Table 1 describes the considered multifractal markers, which are explained at the end of Section 2.2. The paper discusses the experimental evaluations in Section 3 for BreakHis, BACH, and Oral Cancer datasets, and the DCGAN, VAE, NF, and DDPM generative modeling approaches.

**Strengths:**

-The authors' motivation for proposing evaluation scores for generative models in medical image settings is relevant.

**Weaknesses:**

-The definition and measurement of the central quantity $h(q)$ are not clearly specified. From Equation (7) and the surrounding text in Section 2, it remains ambiguous how $h(q)$ is computed in practice and how it serves as a measurable fidelity indicator. Although Table 1 lists several “multifractal markers,” it is difficult to see how these are concretely used to evaluate the fidelity of a generative model. Since the main claim of the paper is to introduce a well-defined evaluation metric, the lack of precision in defining and operationalizing $h(q)$ represents a substantial conceptual weakness.

-The paper does not make it clear in what rigorous or practical sense the proposed multifractal metrics assess the fidelity of generated images. Suppose several generative models are trained on the same dataset, how should one combine or interpret the metrics in Table 1 to *rank* their fidelity performance? The absence of a consistent procedure for aggregating the multifractal scores to rank a group of generative models undermines the claimed utility of the framework.

-The methodology defines $u_{v,w}$ (Eq. 1) directly in pixel space, without using any learned image embeddings or perceptually meaningful feature spaces. It is unclear how such pixel-level quantities can capture perceptual or semantic aspects of image fidelity, or provide robust ranking across different models.

-The experimental section lacks any direct comparison with baseline evaluation scores such as FID, Precision & Recall, or Density & Coverage. Such comparisons are essential to validate that the proposed metrics provide meaningful or improved fidelity assessment over existing methods.

-The authors claim (bottom of page 4) to propose an “information-theoretic framework,” but no standard information-theoretic quantities (like entropy, divergence, or mutual information) appear in the formulation.

**Questions:**

1- What is the precise definition of $h(q)$ in Equation (7)?

2- Do the authors have any comparison to standard fidelity metrics such as FID, Precision, and Density?

3- In what sense, is the authors' proposed evaluation information-theoretic as claimed in Line 100?

---

### Official Review · Reviewer_AwmJ · 2025-11-02

**Soundness:** 2
**Presentation:** 2
**Contribution:** 3
**Rating:** 2
**Confidence:** 4

**Summary:**

The paper introduces multifractal analysis as a method to evaluate generative models, capturing structural complexity and long-range correlations in generated images. Metrics such as the Hurst exponent and singularity spectrum width are proposed and the applications in medical imaging and industrial vision are emphasized.

**Strengths:**

- Novel adaptation of multifractal analysis to GM evaluation.

- Addresses structural and morphological fidelity overlooked by standard metrics.

- Technically thorough and well-motivated for domains requiring fine-grained features.

**Weaknesses:**

Here are my main concerns:

- The paper's core weakness is the insufficient experimental validation. While the idea is novel and interesting, it lacks ablation studies to justify key design choices, such as the selection of hyperparameters (s and q) and to validate the proposed metric itself.

- Beyond the lack of ablation studies, the paper does not provide a direct comparison with established baseline metrics. It would be valuable to see concrete scenarios where baseline metrics fail to accurately assess image quality, while the proposed method succeeds.

- Related works: The paper proposes an information-theoretic framework but overlooks several existing information-theoretic metrics for evaluating generative models, such as RKE [2], Vendi [3], FKEA [4], Ken [5], and FINC [6]. A discussion of how the proposed method relates to these works is important.

- The authors mention the shortcomings of previous evaluation metrics but do not provide experimental comparisons or justification to demonstrate that their proposed method actually improves upon these specific issues.

- The computational cost of the method is not detailed. Furthermore, a convergence analysis is needed to show how many samples are required to obtain reliable and stable metric values.

- Interpretation of multifractal markers in terms of perceptual quality is not fully established. I believe there should be some experiments showing that the method achieved the perceptual quality.

- P1 L051: As suggested by [1] and the future works, these scores can be calculated using more reliable embeddings such as DINOv2, which is not sensitive to ImageNet labels. I believe these discussions need to be added to the paper.


Minor issues:

- Figure 1 visualization can be much improved. Currently, it is a bit hard to read the details.
- P3 L134: it should be _{u,w} (i, j)
- P4 L179: relative relative strength (Additional relative)
- Be careful about the usage of citet and citep. Almost all the citations in the Introduction are citet which should be citep.
- There are no supplementary materials or codes that limit the reproducibility.

---

[1] Stein et. al, "Exposing flaws of generative model evaluation metrics and their unfair treatment of diffusion models", NeurIPS 2023

[2] Jalali et al., “An information-theoretic evaluation of generative models in learning multi-modal distributions”, NeurIPS 2023.

[3] D. Friedman and A. B. Dieng, “The Vendi Score: A Diversity Evaluation Metric for Machine Learning.”, TMLR 2023

[4] Ospanov et al., “Towards a scalable reference-free evaluation of generative models”, NeurIPS 2024.

[5] Zhang et al., "An Interpretable Evaluation of Entropy-based Novelty of Generative Models", ICML 2024

[6] Zhang et al., “Unveiling Differences in Generative Models: A Scalable Differential Clustering Approach”, CVPR 2025.

**Questions:**

- How do you estimate a, b, and c in Eq. 2?

- How would your method work on the general domain of generative models, such as Image (other than medical), or other modalities such as video or text?

- How does Figure 3 show the advantages of your proposed metric?

---

### Meta-Review · Area_Chair_TVq9 · 2026-01-06

**Summary:**

The major concerns are insufficient experimental validation like lack of ablation studies or comparison to previous works, missing of existing works, no analysis of computation cost, lack of technical clarity, unclear or not well supported motivation or claims.

**Reviewer Concerns:**

All three reviewers have quite negative initial reviews for this paper (with score 2, 2, 2). And T=there is no author rebuttal, so the major concerns such as insufficient experimental validation, missing of existing works, no analysis of computation cost, lack of technical clarity are not addressed.

**Reviewer Scores:**

Since there is no rebuttal, the scores should not be changed.

---

### Decision · Program_Chairs · 2026-01-26

Reject